# Shoulder Range of Motion Measurement Using Inertial Measurement Unit–Concurrent Validity and Reliability

**DOI:** 10.3390/s23177499

**Published:** 2023-08-29

**Authors:** Jakub Kaszyński, Cezary Baka, Martyna Białecka, Przemysław Lubiatowski

**Affiliations:** 1Rehasport Clinic, Gorecka 30, 60-201 Poznan, Poland; martyna.bialecka@put.poznan.pl (M.B.); p.lubiatowski@rehasport.pl (P.L.); 2The Faculty of Mechanical Engineering, Institute of Applied Mechanics, Poznan University of Technology, 60-965 Poznan, Poland; 3Orthopaedics, Traumatology and Hand Surgery Department, Poznan University of Medical Sciences, 28 Czerwca 1956, No. 135/147, 61-545 Poznan, Poland

**Keywords:** IMU, wearable movement sensors, shoulder, range of motion, validation

## Abstract

This study aimed to evaluate the reliability of the RSQ Motion sensor and its validity against the Propriometer and electronic goniometer in measuring the active range of motion (ROM) of the shoulder. The study included 15 volunteers (mean age 24.73 ± 3.31) without any clinical symptoms with no history of trauma, disease, or surgery to the upper limb. Four movements were tested: flexion, abduction, external and internal rotation. Validation was assessed in the full range of active shoulder motion. Reliability was revised in full active ROM, a fixed angle of 90 degrees for flexion and abduction, and 45 degrees for internal and external rotation. Each participant was assessed three times: on the first day by both testers and on the second day only by one of the testers. Goniometer and RSQ Motion sensors showed moderate to excellent correlation for all tested movements (ICC 0.61–0.97, LOA < 23 degrees). Analysis of inter-rater reliability showed good to excellent agreement between both testers (ICC 0.74–0.97, LOA 13–35 degrees). Analysis of intra-rater reliability showed moderate to a good agreement (ICC 0.7–0.88, LOA 22–37 degrees). The shoulder internal and external rotation measurement with RSQ Motion sensors is valid and reliable. There is a high level of inter-rater and intra-rater reliability for the RSQ Motion sensors and Propriometer.

## 1. Introduction

The measurement of shoulder range of motion (ROM) is an essential part of the orthopedic examination. It gives the clinician basic but crucial information about the patient’s problem [1]. Currently, using the goniometer for this procedure is a gold standard. This traditional device can measure ROM accurately; however, the results are often influenced by the rater’s knowledge and skills [2,3]. In clinical practice, inclinometer and smartphone applications are commonly used [4,5]. Moreover, optoelectronic systems can assess the quality and quantity of movement comprehensively. Unfortunately, the most significant limitation of those systems is the required laboratory space and the high price of the whole setup [6].

In recent years, wearable motion-tracking systems consisting of a set of Inertial Motion Units (IMU) have become very popular among health providers and sports professionals. Compared with optoelectronic systems, IMUs are light, easily applicable, relatively low cost, and not limited to indoor usage. Therefore, IMU systems can be used in multiple scenarios. A good example is a sports application allowing for ROM assessment in athletes’ natural environment of a football pitch or tennis court to capture human motion and obtain data as precisely as possible [7,8,9,10,11].

The shoulder complex incorporates the glenohumeral, acromioclavicular, sternoclavicular, and scapulothoracic joints, making it the most complicated joint system in the human body [12,13]. The coupled motion of those joints makes it the most challenging part of human motion to capture [8,14].

This problem has to be addressed because of the clinical significance of the proper assessment of shoulder movement in daily practice. In the nearest future, that niche may be filled with fast applicable RSQ Motion sensors.

Past studies which concentrated on the validity of the measurement of shoulder range of motion usually compared readings from IMUs with the gold standard—the goniometer [15,16,17,18]. However, other authors, in their research, compared IMUs with optoelectronic systems as well [19,20,21,22]. Some researchers also underlined that the shoulder is the most challenging joint to assess with IMUs in practice. Our institution has also used and tested the old generation of sensors for joint position sense [17,18]. They were fully wireless and could not assess the motion in a horizontal plane.

This study aimed to evaluate the reliability of the RSQ Motion (RSQ Technologies, Poznań, Poland) sensor and its concurrent validity against the Propriometer (Progres; Ostrów Wielkopolski, Poland) and high-accuracy electronic goniometer in measuring the active range of motion of the shoulder in healthy individuals.

## 2. Materials and Methods

### 2.1. Study Group

The study included 15 volunteers (mean age 24.73 ± 3.31) without any clinical symptoms (including pain) and with no past or present history of trauma, disease, or surgery to the upper limb. Participants’ characteristics are presented in Table 1. All the volunteers were students, and none were past or present professional athletes.

### 2.2. Measurements

Four right and left shoulder movements were tested according to a standardized protocol: flexion, abduction, and external and internal rotation at 90° of shoulder abduction. Flexion and abduction were examined in the sitting position with the spine supported along its entire length so that the scapulas did not come into contact with the wall. External and internal rotation were tested in the supine position with the shoulder abducted to 90°, the elbow flexed to 90°, and the board positioned at a scapular height to stabilize the shoulder girdle. Starting positions are presented in Table 2. ROM was assessed in both shoulders and results from the right and left were pooled.

RSQ Motions sensors were validated against an electronic goniometer and another previously used sensor (Propriometer) [23,24]. Validation (against other devices) was assessed in the full range of active shoulder motion. Reliability was revised in full active ROM, a fixed angle of 90° for flexion and abduction, and 45° for internal and external rotation controlled by goniometers for each direction of motion only against the Propriometer. Two RSQ motion sensors were used simultaneously. One was placed directly on the body part and another was fixed to the Propriometer. The reason for this was to verify the possible impact of IMU alignment differences between compared devices.

Evaluation of range of motion of the shoulder with an electronic goniometer.

An electronic goniometer [CMT, DAF-001; Elbląg, Poland] with a built-in level was used to measure the ROM as a reference. Its measurement range is from 0° to 360° with an accuracy of 0.05°, and the result was shown on the LCD. The examiners (JK or CB) determined the angle based on previously established landmarks (Table 3).

For the evaluation of the flexion angle, the axis of the goniometer’s rotation was at the lateral aspect of the glenohumeral joint, the stationary device’s arm was parallel to the vertical axis, and the moving arm was placed along the participant’s arm, pointing to the lateral humeral epicondyle.

For the abduction measurement, the axis of the goniometer’s rotation was placed at the level of the anterior aspect of the glenohumeral joint, the stationary arm was parallel to the vertical axis, and the moving arm pointed to the lateral humeral epicondyle.

The external and internal rotation with the shoulder abducted to 90° and elbow flexed to 90° were examined by placing the center fulcrum of the goniometer on olecranon, the stationary arm was parallel to the ground, and the moving arm along the participant’s forearm indicated an ulnar styloid.

### 2.3. Evaluation of the Range of Motion of the Shoulder with Sensors

We have used 2 kinds of sensors: Propriometer and RSQMotion.

The Propriometer had been used for joint position sense assessment at our clinic before RSQ Motion sensors replaced it. This device contains a 2-axis accelerometer. The calculated angle is the ratio between data from both axes and the corresponding trigonometric function, which implies the dependence between the vector of the device and vector of gravity, but in a more simplified version than in RSQ Motion. In the Propriometer, neither a gyroscope nor a magnetometer was used, which might have caused higher latency.

RSQ Motion sensor (used as IMU), when put on a chosen body segment, estimates its orientation using a tri-axial gyroscope, tri-axial accelerometer, and magnetometer. The accelerometer is calibrated with the least squares method (LSM) to achieve the desired accuracy. Three parameters are calibrated: scale factor, misalignment, and offset. Madgwick’s motion fusion algorithm (MFA) is used to estimate the orientation of the RSQ Motion sensor with data obtained from the accelerometer and gyroscope. The data from the accelerometer are used to determine the deviation of the motion sensor from the horizontal axis, compensating for the drift error in axes perpendicular to the axis of the gravity vector. On the other hand, the gyroscope data positively minimize the measuring device’s latency. Thus, the combination of an accelerometer and a gyroscope allows for measuring more dynamic motion.

For flexion and abduction, the Propriometer was positioned at the height of the insertion of the deltoid muscle, and the front of the device was halfway between the sagittal and frontal planes. For the external and internal rotation examination, the device was on the dorsal side of the forearm, 2 cm below the ulnar styloid. Two RSQ Motion sensors were used to test the ROM of the shoulder. One (IMU1) was on top of the Propriometer body, and the other one (IMU2) was on the opposite side of the arm. A plastic shim was placed under IMU2 to reduce the impact of the individual anatomical differences, such as muscle mass or adipose tissue. Both sensors were mounted with a Velcro elastic strap.

### 2.4. Procedure

Firstly, the examiner explained the purpose of the experiment and demonstrated and practiced the movement with the participant until it was performed correctly. The IMU sensors were then connected to the handheld devices and calibrated according to the producer’s recommendations. A Propriometer and two IMU sensors were placed on the participant’s arm. Then, each participant was asked to perform full-range, active movement in a chosen direction and hold this position for 3 s. Additionally, ROM was tested in fixed positions, which were set with a goniometer, in 90° of flexion and abduction and 45° in both rotations. One examiner measured the range of motion with the goniometer, and the other one gathered data from the Propriometer, IMU1, and IMU2. To ensure that the range of motion was evaluated simultaneously, a voice command was established during which data was collected.

The first examiner assessed flexion and abduction (Figure 1) with a goniometer then the other examiner repeated measurements in the same manner without changing the sensor alignment. The procedure was then repeated for the opposite arm. The IMU sensors were re-calibrated before the rotation movements were tested. After placing the Propriometer and IMU sensors on the forearm, external and internal rotation measurements were performed with the upper arm abducted to 90° (Figure 2). The researchers made sure that the rotation movements took place only within the glenohumeral joint without compensations from the scapula. After examining each maneuver three times to obtain an average for each examined angle, there was a change of researchers. The trial was repeated for the opposite arm. During each measurement, four angular values were obtained from four different devices for the given movement.

Each participant was assessed three times: on the first day by both testers (to compare the results between raters; inter-rater reliability), on the second day only by one of the testers (to compare the results with those obtained the day before; intra-rater reliability).

Ethical approval was provided by both the internal review and the bioethical Committee of the University of Medical Sciences in Poznań, Poland (no. 883/18). All participants provided informed written consent.

### 2.5. Statistical Analysis

Statistical analysis was performed in Statistica 12. To assess the concurrent validity of active shoulder ROM between RSQ Motion sensors, a Propriometer, and a goniometer for each movement, Intraclass Correlation Coefficient (ICC) and Bland–Altman analysis were used. Furthermore, inter-rater and intra-rater reliability were also assessed by using ICC and Bland–Altman analysis. We included 15 patients in the current study, assessing each arm separately. By doing this we doubled our group to 30 cases. In every given direction each participant performed 3 repetitions. Furthermore, we analyzed data from two days of testing or two raters, which gave us N = 180 cases in analysis of concurrent validity and N = 90 cases in inter- rater and intra-rater analysis. Before conducting the study, we performed power analysis using results that were published by Rigoni et al. [16]. In calculations we used the following assumptions: effect size = 5 degrees, standard deviation = 5 degrees, alpha 0.05, power 0.90. The minimum required sample size was N = 26.

Intra- and inter-rater reliability was tested using a two-way random, single-measures, absolute agreement model ICC (2,1) when comparing to the “gold standard” (the electronic goniometer) or ICC (2,k) when comparing two equivalent devices. According to the guidelines published by Koo and Li [25], the thresholds used to interpret ICC results are shown in Table 4. Bland–Altman plot was used to evaluate the agreement between three different ROM measure instruments by constructing 95% limits of agreement (LOA) for each comparison [26,27]. According to the data reported in the literature, an acceptable difference between measurements requires LOA to be within 10° [16]. However, Mullaney et al. suggested that LOA for shoulder measurements with a goniometer can be acceptable within 15° [28]. Minimal Detectable Change with 95% confidence interval (MDC_95_) and Standard Error of Measurement (SEM) were calculated by following equations:$${\mathrm{MDC}}_{95}\;=\;z{\mathrm{score}}_{\mathrm{Level}\;\mathrm{of}\;\mathrm{confidence}}\;\times \;\sqrt{2}\times \mathrm{SEM}$$
$$\mathrm{SEM}\;=\;{\mathrm{SD}}_{\mathrm{baseline}}\;\times \;\sqrt{1-ICC}$$

MDC95 lower than 10% is considered as excellent, between 10% and 30% is reasonable [29].

The Shapiro–Wilk test was used to assess the normality of the distribution of data. The Wilcoxon test was used for paired data analysis. The level of significance was set to *p* < 0.05.

## 3. Results

### 3.1. Validation of Sensors

The ICC analysis showed a moderate-to-good correlation between the goniometer and IMU sensors in both positions in flexion and abduction with LOA above 20 degrees. Excellent correlation was achieved in internal and external rotation with LOA below 15 degrees. The difference between the goniometer and IMU1 was greater than MDC_95_ in abduction (Table 5 and Table 6).

The goniometer and Propriometer showed good correlation in the abduction and excellent correlation in flexion and internal and external rotation, but an acceptable correlation was observed only in flexion and internal rotation. The difference between the goniometer and Propriometer was greater than MDC_95_ in abduction (Table 7). A comparison of the results of both IMU sensors between each other showed an excellent correlation in all of the examined directions; however, LOA always exceeded 10 degrees. Differences between IMU1 and IMU2 excedeed MDC_95_ in both rotations (Table 8).

A good-to-excellent correlation was observed between a Propriometer and RSQ Motion sensor placed on the Propriometer (IMU1) (Table 9); LOA was greater than 10 degrees only in flexion. On the other hand, a Propriometer and sensor placed directly on the arm (IMU2) showed a moderate correlation in flexion, good in abduction, and excellent in both rotations, but LOA exceeded 10 degrees in every movement. Additionally, differences in both rotations were greater than MDC_95_ (Table 10).

### 3.2. Inter-Rater Reliability

Good to excellent agreements between both testers (T1 and T2) were observed for each device in all of the examined movements (except internal rotation): goniometer: ICC 0.88–0.96; IMU1: ICC 0.93–0.94; IMU2: ICC 0.88–0.97; Propriometer: ICC 0.94–0.97. Moreover, LOA did not exceed 19 degrees in every device’s given directions (Table 11, Table 12, Table 13 and Table 14).

Relatively poor correlation with wide LOA was noted in internal rotation for IMU sensors in both locations (IMU 1: ICC 0.77; IMU 2: ICC 0.74; LOA 35 degrees for both), although the difference between raters remained unchanged. Furthermore, the highest values of MDC_95_ were observed in internal rotation (Table 14).

The results of the analysis of fixed shoulder positions set with the goniometer (90 degrees in flexion and abduction, 45 degrees in internal and external rotation) were ambiguous (Table 15, Table 16, Table 17 and Table 18).

In flexion, abduction, and internal rotation, the most accurate indications were seen on the Propriometer and IMU1 (max. reported difference: 2 degrees; max. reported LOA: 11 degrees). In those movements, readings showed by sensor IMU2, placed directly on the arm, were 5–7 degrees away from the goniometer.

In external rotation, none of the used devices showed agreement with the goniometer. Nevertheless, inter-rater reliability remained very good for each device, with differences between testers within 2 degrees and LOA up to only 10 degrees.

The Willcoxon test was performed to assess whether there is a significant difference between angles set with the goniometer (90 or 45 degrees) and other sensors’ readings. There was no significant difference between the goniometer and IMU1 in flexion for both testers and Propriometer for tester 2.

### 3.3. Intra-Rater Reliability

Intra-rater reliability was good for the goniometer, IMU1, and the Propriometer in every examined direction (ICC 0.79–0.88), but LOA was always above 20 degrees. Agreements for IMU2 were moderate to good (ICC 0.7–0.84; LOA 28 degrees). However, external rotation was the most challenging movement for all of the devices used in this study. In this direction, the differences between day 1 and day 2 ranged from 3 ± 8 to 5 ± 8 degrees and LOA from 30 to 37 degrees (Table 19, Table 20, Table 21 and Table 22).

In 90 degrees of flexion and abduction, IMU1 and Propriometer were more accurate than IMU2 (Table 23 and Table 24). In 45 degrees of internal rotation, IMU1 and Propriometer were superior to IMU2 (Table 25). However, in external rotation, the Propriometer showed the poorest result in comparison to the angle set with the goniometer (Table 26). LOA for all given directions and devices ranged from 7 to 16 degrees.

In the Willcoxon test, there was no significant difference between the goniometer and IMU1 in flexion on Day 1 and Day 2 and Propriometer on Day 1.

## 4. Discussion

Validation of the RSQMotion sensors for internal and external rotation achieved excellent scores compared to reference measurements and between sensors. Validation for flexion and abduction showed good correlations; however, agreements were not acceptable. Nevertheless, inter-rater and intra-rater indicators were proven to have good-to-excellent agreement regardless of the position of IMU.

Although the literature is not extensive in studies focusing on the validation of IMU sensors vs. goniometer in measuring ROM of the shoulder joint, we were able to find some similarities to some of the works. As well as in our research, Rigoni et al. [16] and Bravi et al. [17] observed high ICC values between devices (Table 27). However, Rigoni et al. reported much smaller ranges of LOA (−4.5° to 3.2°). In contrast, in the study by Bravi et al., the calculated LOA was always greater than 15°.

In the studies mentioned above [16,17], the reported maximum ROM of the shoulder among the healthy group was much lower than in our study (Table 27). This might result in a more negligible difference between devices or raters as well.

When examining static positions at the predetermined 90° in flexion and abduction and 45° in internal rotation, the obtained results were most accurate using the Propriometer and IMU1 sensor. In external rotation, none of the sensors agreed with the goniometer. Yoon [15] also noted in his study that the results in 45° of external rotation are not accurate (the 95% LOA for the discrepancy between the measurements exceeded ± 5°).

A goniometer has been a standard tool to assess the ROM of any joint in the physiotherapeutic or orthopedic room. However, the most significant limitation of this device is that the obtained results often depend on the physician’s skills and experience and their engagement [2,3]. On the other hand, the application of the device itself and data acquisition make the RSQ Motion sensor very easy to use (user friendly). That does not mean that shoulder motion evaluation is an easy task, due to complex movements, involvement of scapula, spine, and glenohumeral joint itself. In this study, we showed that the RSQ Motion sensor might replace the goniometer in measuring internal and external rotation. However, the assessment of shoulder flexion and abduction still needs some improvement. One of the possible solutions is the placement of an RSQ Motion sensor on the forearm because the elbow stays extended during movements. Of course, this idea needs further research to evaluate the validity and reliability of this procedure.

Considering intra-rater reliability, the only moderate-to-good correlation between the RSQ Motion sensor and goniometer with high MDC_95_ was observed in internal rotation. During measurements, we observed that this movement was the most difficult for participants because some of them could not control the scapula properly, limiting the motion only to the glenohumeral joint. We believe that if examiners helped stabilize subjects’ scapula, the correlation would be higher.

In our study, we calculated the MDC_95_ to address the issue of responsiveness and change detection potential in our measurements. The MDC_95_ is a crucial statistical parameter that quantifies the smallest detectable difference between two measurements within a group of patients. It helps us to determine whether the observed changes in our study are not merely due to measurement error or random fluctuations. According to the work of Kaszyński et al. [30], MDC_95_, also known as the smallest real change, is a statistical estimate of a value that can be detected by a measurement. It reflects changes that fall outside the SEM of a given test. Changes exceeding the MDC_95_ value are considered clinically relevant. In the current study, such differences were observed only in the comparison of different devices with each other: goniometer and IMU1 in abduction, IMU1 and IMU2 in flexion, internal and external rotation, Propriometer and IMU2 in internal and external rotation. Furthermore, we observed that MDC_95_ was greater than 10% in internal rotation in comparison of both testers and in both rotations for every tested device in day-to-day comparison (intra-rater reliability). However, all of the calculated MDC_95_ did not exceed 30%. 

It seems that the arrangement of the sensors on the body of the examined person had the greatest influence on the repeatability of the measurements. The patient-to-patient differences in the shape and size of the muscles and the amount of subcutaneous adipose tissue make it very difficult to apply the sensor in the same position for each examined person, despite the previously established reference point. To reduce the influence of the body shape on the test results, a pad was used, which was placed under the IMU sensor. After this change, we observed an improvement in the reproducibility of the results. However, it seems that modifications should be made to reduce further the influence of the body curvature on the results obtained. Another option we currently verify is using a whole system of motion capture sensors to obtain the readings from two or three sensors representing the body parts and using specifically designed algorithms also using machine learning.

Optoelectronic systems consist of a set of markers placed on precisely described spots on the human body, which are tracked by cameras. It allows to analyze of human joint kinematics with a high measurement accuracy of 0.1 mm in a position [31]. That is why they are often considered a laboratory gold standard in motion capture [32]. In the available research, which focused on the validity of measurement of the range of motion of the upper limb with those systems, it has been shown that there is a high correlation coefficient with very low LOA compared to a standard goniometer [33]. Those results were better than those presented in the current study and the other studies determining inter- and intra-rater reliability of IMU systems in shoulder assessment [16,17].

Optoelectronic systems are currently considered to be the “gold standard” in motion capture analysis. IMU seems to approaching an equivalent value in many terms with a much lower cost, accessibility, and no need for a dedicated laboratory and specialized personnel.

The RSQ Motion sensors were first tested in laboratory conditions on the KUKA robot for geometric indications. Their accuracy and repeatability are excellent, so it should be assumed that the device works flawlessly and its readings did not affect the obtained measurement differences in the final test result [34].

Undoubtedly, the strongest point of this study is an evaluation of intra-rater reliability. We assumed that no changes in shoulder ROM could occur within two testing days. Thus, thanks to this protocol, we proved that the repeatability of measurements taken with RSQ Motion is at a very high level.

This study also has some limitations. Firstly, our experimental group consisted of only healthy participants without past shoulder injuries, which can imply some differences in measurements in patients with shoulder-related problems. The second limitation is the small sample size in this study; however, we analyzed each shoulder separately to achieve the optimal amount of data. The examination took approximately 30 min, and each patient was tested twice. We cannot exclude the influence of the psychophysical fatigue of the participants on the obtained data. It is known that the obtained results may be influenced by the magnetic field of the building in which the test took place. We have not tested it and we cannot exclude its influence, but each test was carried out in the same place and was preceded by sensor calibration, so we assume that this influence was insignificant.

During each test, the same measurement conditions were reproduced, uniform voice commands were used, and the participants’ attention was paid to being focused and precise. None of the participants complained of discomfort during the testing. No adverse effects were noted throughout the study.

## 5. Conclusions

The measurement of the shoulder’s internal and external rotation with RSQ Motion sensors is valid and reliable. On the other hand, the assessments of flexion and abduction, although showing a good correlation, still need some protocol changes or a different sensor placement to achieve the desired level of reliability. There is a high level of inter-rater and intra-rater reliability for the RSQ Motion sensors and Propriometer.

## Figures and Tables

**Figure 1 sensors-23-07499-f001:**
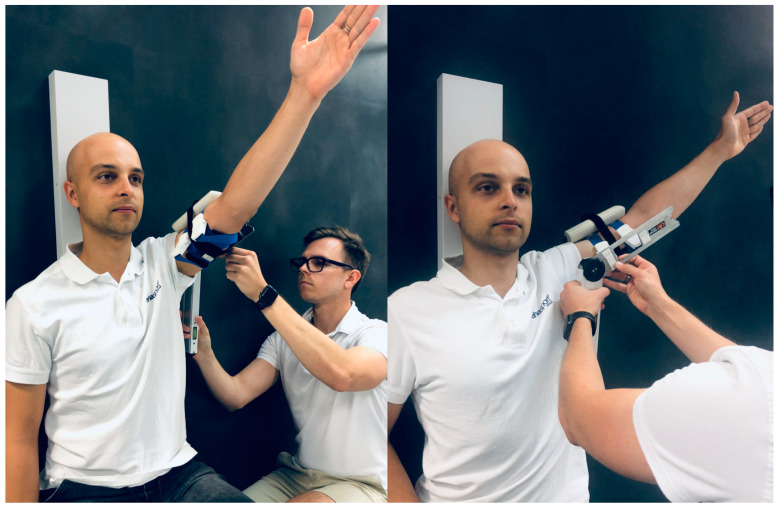
The measurement of flexion (**left**) and abduction (**right**) of the shoulder.

**Figure 2 sensors-23-07499-f002:**
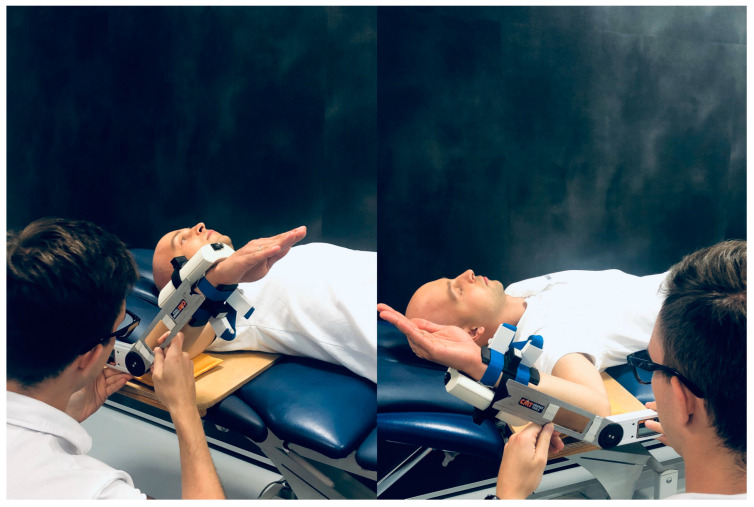
The measurement of internal (**left**) and external (**right**) rotation of the shoulder.

**Table 1 sensors-23-07499-t001:** Participant demographics.

Number	Gender (n)	Age * (Years)	Right Hand Dominant (n)	BMI *	Past Shoulder Injuries (n)
15	F = 5 M = 10	24.7 ± 3.3	15 (100%)	22.3 ± 2.5	0

* data presented with an average ± standard deviation; n-number of participants.

**Table 2 sensors-23-07499-t002:** Starting shoulder positions for ROM measurements.

Movement	Position	Shoulder	Elbow	Forearm
Flexion	Sitting	0°	0°	-
Abduction	Sitting	0°	0°	-
Internal rotation	Lying in supine	90° abduction	90°	Vertical
External rotation	Lying in supine	90° abduction	90°	Vertical

**Table 3 sensors-23-07499-t003:** Goniometer landmarks at shoulder ROM measurements.

Movement	Axis	Stationary Arm	Moving Arm
Flexion	Anterior aspect on acromion	Vertical axis	Pointing to lateral humeral epicondyle
Abduction	Anterior aspect on acromion	Vertical axis	Pointing to lateral humeral epicondyle
Internal rotation	Olecranon	Horizontal axis	Pointing to ulnar styloid
External rotation	Olecranon	Horizontal axis	Pointing to ulnar styloid

**Table 4 sensors-23-07499-t004:** Interpretation for ICC agreement measures [25].

Reliability	ICC
Poor	<0.5
Moderate	0.50–0.74
Good	0.75–0.9
Excellent	>0.9

ICC—Intra-Class Correlation coefficient.

**Table 5 sensors-23-07499-t005:** Comparison of the results between goniometer and RSQ Motion sensor placed on Propriometer (IMU1) in the measurement of maximal range of motion of flexion, abduction, and internal and external rotation. N = 180.

	Goniometer [°]	IMU 1 [°]	Difference (G-IMU1) [°]	ICC (95% Cl)	LOA [°]	MDC_95_ (%)
FLX	170 ± 7	161 ± 8	9 ± 5	0.74 (0.68–0.79)	−2; 19	10.6 (6.4%)
ABD	173 ± 7	162 ± 8	11 ± 6	0.75 (0.69–0.8)	1; 22	10.4 (6.2%)
InR	65 ± 10	67 ± 10	2 ± 3	0.97 (0.96–0.98)	−7; 3	4.8 (7.3%)
ExR	91 ± 11	88 ± 13	3 ± 3	0.96 (0.95–0.97)	−4; 10	6.7 (7.4%)

ICC—Intra-Class Correlation coefficient; 95% CI—95% Confidence Interval; LOA—Limits of Agreement; MDC_95_—Minimal Detectable Change with 95% confidence interval.

**Table 6 sensors-23-07499-t006:** Comparison of the results between goniometer and RSQ Motion sensor placed directly on the arm (IMU2) in the measurement of maximal range of motion of flexion, abduction, and internal and external rotation. N = 180.

	Goniometer [°]	IMU 2 [°]	Difference (G-IMU2) [°]	ICC (95% Cl)	LOA [°]	MDC_95_ (%)
FLX	170 ± 7	168 ± 6	2 ± 6	0.68 (0.6–0.74)	−8; 13	10.2 (6.0%)
ABD	173 ± 7	167 ± 6	6 ± 6	0.61 (0.53–0.69)	−5; 17	11.3 (6.6%)
InR	65 ± 10	59 ± 10	6 ± 3	0.95 (0.94–0.97)	0; 12	6.2 (10.0%)
ExR	91 ± 11	94 ± 12	3 ± 3	0.96 (0.95–0.97)	−9; 3	6.4 (6.9%)

ICC—Intra-Class Correlation coefficient; 95% CI—95% Confidence Interval; LOA—Limits of Agreement; MDC_95_—Minimal Detectable Change with 95% confidence interval.

**Table 7 sensors-23-07499-t007:** Comparison of the results between the goniometer and Propriometer in the measurement of maximal range of motion of flexion, abduction, and internal and external rotation. N = 180.

	Goniometer [°]	Propriometer [°]	Difference (G-P) [°]	ICC (95% Cl)	LOA [°]	MDC_95_ (%)
FLX	170 ± 7	166 ± 8	4 ± 3	0.95 (0.94–0.96)	−2; 10	4.7 (2.8%)
ABD	173 ± 7	164 ± 8	9 ± 6	0.82 (0.77–0.86)	−2; 21	8.8 (5.2%)
InR	65 ± 10	67 ± 11	2 ± 3	0.98 (0.98–0.99)	−7; 3	4.1 (6.2%)
ExR	91 ± 11	86 ± 14	5 ± 4	0.97 (0.96–0.98)	−4; 13	6.0 (6.8%)

ICC—Intra-Class Correlation coefficient; 95% CI—95% Confidence Interval; LOA—Limits of Agreement; MDC_95_—Minimal Detectable Change with 95% confidence interval.

**Table 8 sensors-23-07499-t008:** Comparison of the results between the RSQ Motion sensor placed on Propriometer (IMU1) and RSQ Motion sensor placed directly on the arm (IMU2) in the measurement of maximal range of motion of flexion, abduction, internal and external rotation. N = 180.

	IMU 1 [°]	IMU 2 [°]	Difference (IMU1-IMU2) [°]	ICC (95% Cl)	LOA [°]	MDC_95_ (%)
FLX	161 ± 8	168 ± 6	7 ± 4	0.93 (0.91–0.94)	−14; 1	5.1 (3.1%)
ABD	162 ± 8	167 ± 6	5 ± 4	0.91 (0.89–0.93)	−13; 3	5.8 (3.5%)
InR	67 ± 10	59 ± 10	8 ± 2	0.99 (0.98–0.99)	4; 13	2.8 (4.4%)
ExR	88 ± 13	94 ± 12	6 ± 3	0.99 (0.98–0.99)	−12; −1	3.5 (3.8%)

ICC—Intra-Class Correlation coefficient; 95% CI—95% Confidence Interval; LOA—Limits of Agreement; MDC_95_—Minimal Detectable Change with 95% confidence interval.

**Table 9 sensors-23-07499-t009:** Comparison of the results between the Propriometer and RSQ Motion sensor placed on Propriometer (IMU1) in the measurement of maximal range of motion of flexion, abduction, and internal and external rotation. N = 180.

	Propriometr [°]	IMU 1 [°]	Difference (P-IMU1) [°]	ICC (95% Cl)	LOA [°]	MDC_95_ (%)
FLX	166 ± 8	161 ± 8	5 ± 4	0.84 (0.8–0.87)	−4; 14	8.9 (5.4%)
ABD	164 ± 8	162 ± 8	2 ± 2	0.96 (0.95–0.97)	−3; 6	4.4 (2.7%)
InR	67 ± 11	67 ± 10	0 ± 1	0.99 (0.99–0.99)	−2; 3	2.9 (4.3%)
ExR	86 ± 14	88 ± 13	−1 ± 2	0.99 (0.99–0.99)	−5; 2	3.7 (4.3%)

ICC—Intra-Class Correlation coefficient; 95% CI—95% Confidence Interval; LOA—Limits of Agreement; MDC_95_—Minimal Detectable Change with 95% confidence interval.

**Table 10 sensors-23-07499-t010:** Comparison of the results between the Propriometer and RSQ Motion sensor placed directly on the arm (IMU2) in the measurement of maximal range of motion of flexion, abduction, and internal and external rotation. N = 180.

	Propriometr [°]	IMU 2 [°]	Difference (P-IMU2) [°]	ICC (95% Cl)	LOA [°]	MDC_95_
FLX	166 ± 8	168 ± 6	−1.5 ± 6	0.66 (0.59–0.73)	−13; 10	11.3 (6.8%)
ABD	164 ± 8	167 ± 6	−3 ± 5	0.77 (0.72–0.82)	−13; 7	9.3 (5.6%)
InR	67 ± 11	59 ± 10	8 ± 3	0.96 (0.95–0.97)	3; 14	5.8 (9.2%)
ExR	86 ± 14	94 ± 12	−8 ± 3	0.96 (0.95–0.97)	−15; −1	7.2 (8.0%)

ICC—Intra Class Correlation coefficient; 95% CI—95% Confidence Interval; LOA—Limits of Agreement; MDC_95_—Minimal Detectable Change with 95% confidence interval.

**Table 11 sensors-23-07499-t011:** Inter-rater reliability indicators in shoulder flexion for all of the 4 measuring devices; Tester 1 vs. Tester 2. N = 90.

FLX	Tester 1 [°]	Tester 2 [°]	Difference (T1-T2) [°]	ICC (95% Cl)	LOA [°]	MDC_95_
Goniometer	171 ± 7	170 ± 8	1 ± 3	0.95 (0.92–0.96)	−6; 7	4.7 (2.7%)
IMU1	161 ± 8	163 ± 8	2 ± 4	0.93 (0.88–0.95)	−9; 6	5.9 (3.6%)
IMU2	168 ± 7	168 ± 6	0 ± 3	0.92 (0.89–0.94)	−7; 6	5.1 (3.0%)
Propriometer	166 ± 8	166 ± 8	0 ± 4	0.95 (0.93–0.96)	−7; 7	5.0 (3.0%)

ICC—Intra-Class Correlation coefficient; 95% CI—95% Confidence Interval; LOA—Limits of Agreement; MDC_95_—Minimal Detectable Change with 95% confidence interval.

**Table 12 sensors-23-07499-t012:** Inter-rater reliability indicators in shoulder abduction for all of the 4 measuring devices; Tester 1 vs. Tester 2. N = 90.

ABD	Tester 1 [°]	Tester 2 [°]	Difference (T1-T2) [°]	ICC (95% Cl)	LOA [°]	MDC_95_
Goniometer	173 ± 6	173 ± 7	0 ± 4	0.88 (0.83–0.92)	−9; 8	6.2 (3.6%)
IMU1	162 ± 8	162 ± 9	0 ± 4	0.94 (0.91–0.96)	−8; 8	5.8 (3.6%)
IMU2	167 ± 6	167 ± 7	0 ± 4	0.88 (0.83–0.91)	−8; 9	6.2 (3.7%)
Propriometer	164 ± 8	164 ± 9	0 ± 4	0.94 (0.92–0.96)	−8; 7	5.8 (3.5%)

ICC—Intra-Class Correlation coefficient; 95% CI—95% Confidence Interval; LOA—Limits of Agreement; MDC_95_—Minimal Detectable Change with 95% confidence interval.

**Table 13 sensors-23-07499-t013:** Inter-rater reliability indicators in shoulder external rotation for all of the 4 measuring devices; Tester 1 vs. Tester 2. N = 90.

ExR	Tester 1 [°]	Tester 2 [°]	Difference (T1-T2) [°]	ICC (95% Cl)	LOA [°]	MDC_95_
Goniometer	92 ± 11	91 ± 11	1 ± 4	0.96 (0.93–0.97)	−7; 10	6.1 (6.7%)
IMU1	88 ± 13	87 ± 13	1 ± 5	0.97 (0.95–0.98)	−8; 10	6.2 (7.1%)
IMU2	95 ± 12	94 ± 12	1 ± 4	0.97 (0.95–0.98)	−8; 9	5.8 (6.1%)
Propriometer	87 ± 13	86 ± 14	1 ± 4	0.97 (0.95–0.98)	−9; 10	6.5 (7.5%)

ICC—Intra-Class Correlation coefficient; 95% CI—95% Confidence Interval; LOA—Limits of Agreement; MDC_95_—Minimal Detectable Change with 95% confidence interval.

**Table 14 sensors-23-07499-t014:** Inter-rater reliability indicators in shoulder internal rotation for all of the 4 measuring devices; Tester 1 vs. Tester 2. N = 90.

InR	Tester 1 [°]	Tester 2 [°]	Difference (T1-T2) [°]	ICC (95% Cl)	LOA [°]	MDC_95_
Goniometer	65 ± 10	66 ± 10	1 ± 5	0.94 (0.91–0.96)	−10; 8	6.8 (10.4%)
IMU1	67 ± 10	67 ± 10	0 ± 9	0.77 (0.68–0.84)	−17; 18	13.3 (19.8%)
IMU2	59 ± 10	59 ± 10	0 ± 9	0.74 (0.63–0.82)	−17; 18	14.1 (24.0%)
Propriometer	67 ± 10	67 ± 11	0 ± 7	0.88 (0.82–0.91)	−13; 14	10.1 (15.1%)

ICC—Intra-Class Correlation Coefficient; 95% CI—95% Confidence Interval; LOA—Limits of Agreement; MDC_95_—Minimal Detectable Change with 95% confidence interval.

**Table 15 sensors-23-07499-t015:** Comparison of results of IMU1, IMU2, and Propriometer in 90 degrees of shoulder flexion; Tester 1 vs. Tester 2. N = 90.

FLX 90°	Tester 1 [°]	Tester 2 [°]	Difference (T1-T2) [°]	LOA
IMU1	90 ± 2	90 ± 2	0 ± 2	−3; 3
IMU2	95 ± 3 *	95 ± 2 *	0 ± 2	−3; 4
Propriometer	90 ± 2 *	90 ± 2	0 ± 2	−3; 4

* significant difference between goniometer and given device; LOA—Limits of Agreement.

**Table 16 sensors-23-07499-t016:** Comparison of results of IMU1, IMU2, and Propriometer in 90 degrees of shoulder abduction; Tester 1 vs. Tester 2. N = 90.

ABD 90°	Tester 1 [°]	Tester 2 [°]	Difference (T1-T2) [°]	LOA
IMU1	91 ± 2 *	91 ± 1 *	0 ± 2	−3; 3
IMU2	96 ± 2 *	96 ± 2 *	0 ± 2	−4; 4
Propriometer	91 ± 2 *	91 ± 2 *	0 ± 2	−4; 4

* significant difference between goniometer and given device; LOA—Limits of Agreement.

**Table 17 sensors-23-07499-t017:** Comparison of results of IMU1, IMU2, and Propriometer in 45 degrees of shoulder internal rotation; Tester 1 vs. Tester 2. N = 90.

InR 45°	Tester 1 [°]	Tester 2 [°]	Difference (T1-T2) [°]	LOA
IMU1	47 ± 1 *	47 ± 2 *	0 ± 2	−5; 4
IMU2	38 ± 3 *	38 ± 3 *	0 ± 3	−6; 5
Propriometer	47 ± 1 *	47 ± 2 *	0 ± 3	−5; 5

* significant difference between goniometer and given device; LOA—Limits of Agreement.

**Table 18 sensors-23-07499-t018:** Comparison of results of IMU1, IMU2, and Propriometer in 45 degrees of shoulder external rotation; Tester 1 vs. Tester 2. N = 90.

ExR 45°	Tester 1 [°]	Tester 2 [°]	Difference (T1-T2) [°]	LOA
IMU1	41 ± 3 *	43 ± 3 *	2 ± 2	−7; 3
IMU2	48 ± 3 *	50 ± 3 *	2 ± 2	−7; 2
Propriometer	39 ± 3 *	41 ± 4 *	2 ± 2	−7; 3

* significant difference between goniometer and given device; LOA—Limits of Agreement.

**Table 19 sensors-23-07499-t019:** Intra-rater reliability indicators for goniometer; Day 1 vs. Day 2. N = 90.

Goniometer	Day 1 [°]	Day 2 [°]	Difference (D1-D2) [°]	ICC (95% Cl)	LOA [°]	MDC_95_
FLX	170 ± 7	170 ± 9	0 ± 6	0.86 (0.8–0.9)	−11; 12	8.3 (4.9%)
ABD	173 ± 6	172 ± 7	1 ± 5	0.8 (0.71–0.86)	−10; 11	8.1 (4.7%)
InR	65 ± 10	65 ± 12	0 ± 9	0.8 (0.72–0.86)	−17; 18	13.6 (21.0%)
ExR	92 ± 11	89 ± 12	3 ± 8	0.86 (0.77–0.9)	−12; 18	11.9 (13.2%)

ICC—Intra-Class Correlation coefficient; 95% CI—95% Confidence Interval; LOA—Limits of Agreement; MDC_95_—Minimal Detectable Change with 95% confidence interval.

**Table 20 sensors-23-07499-t020:** Intra-rater reliability indicators for RSQ Motion sensor placed on Propriometer (IMU1); Day 1 vs. Day 2. N = 90.

IMU 1	Day 1 [°]	Day 2 [°]	Difference (D1-D2) [°]	ICC (95% Cl)	LOA [°]	MDC_95_
FLX	161 ± 8	163 ± 9	2 ± 7	0.8 (0.71–0.86)	−15; 12	10.5 (6.5%)
ABD	162 ± 8	162 ± 9	0 ± 6	0.88 (0.83–0.92)	−11; 11	8.2 (5.0%)
InR	67 ± 10	67 ± 11	0 ± 8	0.83 (0.76–0.88)	−16; 17	12.0 (17.9%)
ExR	89 ± 13	86 ± 13	3 ± 9	0.82 (0.73–0.88)	−15; 22	15.3 (17.5%)

ICC—Intra-Class Correlation coefficient; 95% CI—95% Confidence Interval; LOA—Limits of Agreement; MDC_95_—Minimal Detectable Change with 95% confidence interval.

**Table 21 sensors-23-07499-t021:** Intra-rater reliability indicators for RSQ Motion sensor placed directly on the arm (IMU2); Day 1 vs. Day 2. N = 90.

IMU 2	Day 1 [°]	Day 2 [°]	Difference (D1-D2) [°]	ICC (95% Cl)	LOA [°]	MDC_95_
FLX	168 ± 7	168 ± 8	0 ± 7	0.7 (0.57–0.79)	−14; 14	11.4 (6.8%)
ABD	167 ± 6	167 ± 8	0 ± 6	0.79 (0.7–0.85)	−11; 12	8.9 (5.3%)
InR	59 ± 10	60 ± 11	1 ± 8	0.83 (0.76–0.88)	−17; 15	12.0 (20.2%)
ExR	95 ± 13	90 ± 13	5 ± 8	0.84 (0.7–0.9)	−11; 20	14.4 (15.6%)

ICC—Intra-Class Correlation coefficient; 95% CI—95% Confidence Interval; LOA—Limits of Agreement; MDC_95_—Minimal Detectable Change with 95% confidence interval.

**Table 22 sensors-23-07499-t022:** Intra-rater reliability indicators for Propriometer; Day 1 vs. Day 2. N = 90.

Propriometer	Day 1 [°]	Day 2 [°]	Difference (D1-D2) [°]	ICC (95% Cl)	LOA [°]	MDC_95_
FLX	166 ± 8	166 ± 9	0 ± 6	0.82 (0.75–0.88)	−13; 12	10.0 (6.0%)
ABD	163 ± 8	163 ± 8	0 ± 5	0.88 (0.83–0.92)	−10; 11	7.7 (4.7%)
InR	68 ± 11	67 ± 11	1 ± 8	0.85 (0.79–0.9)	−15; 17	11.8 (17.5%)
ExR	87 ± 14	84 ± 13	3 ± 10	0.83 (0.75–0.89)	−15; 22	15.4 (18.1%)

ICC—Intra Class Correlation coefficient; 95% CI—95% Confidence Interval; LOA—Limits of Agreement; MDC_95_—Minimal Detectable Change with 95% confidence interval.

**Table 23 sensors-23-07499-t023:** Comparison of results of IMU1, IMU2, and Propriometer in 90 degrees of shoulder flexion; Day 1 vs. Day 2. N = 90.

FLX 90°	Day 1 [°]	Day 2 [°]	Difference (D1-D2) [°]	LOA [°]
IMU1	90 ± 2	90 ± 2	0 ± 1	−6; 5
IMU2	96 ± 3 *	95 ± 4 *	1 ± 1	−6; 8
Propriometer	90 ± 2	89 ± 2 *	1 ± 0	−3; 5

* significant difference between goniometer and given device; LOA—Limits of Agreement.

**Table 24 sensors-23-07499-t024:** Comparison of results of IMU1, IMU2, and Propriometer in 90 degrees of shoulder abduction; Day 1 vs. Day 2. N = 90.

ABD 90°	Day 1 [°]	Day 2 [°]	Difference (D1-D2) [°]	LOA [°]
IMU1	91 ± 1 *	92 ± 2 *	1 ± 1	−4; 3
IMU2	98 ± 3 *	98 ± 3 *	0 ± 1	−6; 7
Propriometer	92 ± 2 *	91 ± 2 *	1 ± 0	−3; 5

* significant difference between goniometer and given device; LOA—Limits of Agreement.

**Table 25 sensors-23-07499-t025:** Comparison of results of IMU1, IMU2 and Propriometer in 45 degrees of shoulder internal rotation; Day 1 vs. Day 2. N = 90.

InR 45°	Day 1 [°]	Day 2 [°]	Difference (D1-D2) [°]	LOA [°]
IMU1	47 ± 2 *	46 ± 2 *	1 ± 1	−4; 6
IMU2	39 ± 3 *	39 ± 4 *	0 ± 1	−9; 7
Propriometer	48 ± 2 *	47 ± 2 *	1 ± 0	−3; 6

* significant difference between goniometer and given device; LOA—Limits of Agreement.

**Table 26 sensors-23-07499-t026:** Comparison of results of IMU1, IMU2, and Propriometer in 45 degrees of shoulder external rotation; Day 1 vs. Day 2. N = 90.

ExR 45°	Day 1 [°]	Day 2 [°]	Difference (D1-D2) [°]	LOA [°]
IMU1	41 ± 3 *	42 ± 2 *	1 ± 1	−6; 5
IMU2	49 ± 3 *	47 ± 2 *	2 ± 1	−5; 8
Propriometer	39 ± 3 *	39 ± 3 *	0 ± 1	−6; 5

* significant difference between goniometer and given device; LOA—Limits of Agreement.

**Table 27 sensors-23-07499-t027:** Comparison of active ROM of the shoulder between studies.

	Rigoni et al. [16]		Bravi et al. [17]		Current Study	
	IMU [°]	Goniometer [°]	IMU [°]	Goniometer [°]	IMU [°]	Goniometer [°]
Flexion	155 ± 14	155 ± 15	134 ± 20	140 ± 18	161 ± 8	170 ± 7
Abduction	152 ± 18	151 ± 19	149 ± 21	146 ± 20	162 ± 8	173 ± 7
Internal rotation	53 ± 17	52 ± 18	56 ± 13	53 ± 14	67 ± 10	65 ± 10
External rotation	90 ± 17	89 ± 18	78 ± 18	79 ± 16	88 ± 13	91 ± 11

## Data Availability

Not applicable.

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
