# Peer review of "Shoulder Range of Motion Measurement Using Inertial Measurement Unit–Concurrent Validity and Reliability"

_sensors, 2023, doi:10.3390/s23177499_

Round 1

Reviewer 1 Report

In this paper, the authors said they aimed to evaluate the reliability of the RSQ Motion sensor and its validity against another sensor and electronic goniometer in measuring the active range of motion (ROM) of the shoulder. It was concluded that the shoulder internal and external rotation measurement with RSQ Motion sensors is valid and reliable. However, some details of the manuscript need to be refined. On this basis, a major modification is required before the publication can be accepted.

1. I suggest visualizing the comparison results in the table section to increase the readability of the article.

2. In "Abstract" section," This study aimed to evaluate the reliability of the RSQ Motion sensor and its validity against another sensor and electronic goniometer in measuring the active range of motion (ROM) of the shoulder. this sentence is poorly readable, I suggest indicates the specific kind of "another sensor".

3. This paper has a single testing algorithm, and I suggest that multiple algorithms can be used for testing to increase the confidence of the results.

4. The language skill of all this paper should improved.

1. The language skill of all this paper should be improved.

Reviewer 2 Report

The manuscript entitled "Shoulder Range of Motion Measurement Using Inertial Measurement Unit – Clinical Validity and Reliability" Aims to assess the reliability and validity of an inertial measurement unit in shoulder range of motion measurement. The study is in general quite clear and correctly performed. However several methological issues as well as the main research question render it hardly acceptable. The main concern is that the manuscript does not clearly states the advantages of the newly proposed system in daily clinical practice.

Please find below detailed comments

Title

pg 1 line 1-3 Authors should be more specific when describing validity. The term "clinical validity" is not correct and instead should changed by "concurrent" or "criterion" validity, which is in fact what the authors are testing.

Abstract

Pg 1 line 9 Authors should explain the meaning of RSQ.

Introduction

Pg 2 line 49 Authors should add the meaning of RSQ. 

Pg 2 lines 57-59 the type of validity tested should be clearly specified.

Materials and Methods

In this Sections the numbering is not clear.

I would propose.

2.1 Study group or better "participants"

2.2. (and not 3) Measurements (add the term "gold standard"?) 

2.3. (and not 4) Evaluation of range of motion of the shoulder with sensors.

2.4. Procedure

2.5 Statistical Analysis (pg 5 line 170)

Pg 2 lines 77-78 Authors should explain why they pooled right and left side measurements.

Pg 5 lines 170-174 In these lines the authors are describing validity assessment not "agreement" thus they should remake the paragraph accordingly.

Pg 6 line 176, the term reliability is more appropriate than agreement in this line.

Pg 6 lines 181-182 Authors should provide a reference supporting the statement. 

Results 

Pg 9 lines 258-260 What do the authors exactly mean by the term "ambiguous"?

Discussion

Pg 13 lines 355-358 Authors state that the RSQ motion sensors are "easy to use". However, the methods section mentions several methodological issues (i.e. scapular stabilization) before their use. A clarification is needed since the procedure does not look that easy. 

Pg 13 lines 358-360 Authors should explain better the advantages of a system that only shows acceptable results in rotation measurements.

Pg 14 lines 371-382 This paragraph has a great deal of speculative statements. I think it could be supressed.

Pg 14 lines 392 - 398 The main objective of the study is comparison of inertial sensors to conventional goniometry. Thus the comparison to motion analysis systems is not included. I find that mentioning the disadvantages of motion analysis systems is irrelevant in this study and could be suppressed. 

Pg 14 lines 399-407 this paragraph does not add any relevant information in present study and should be suppressed.

Pg 14 lines 408-411 is a possible limitation and should be mentioned accordingly.

Pg 14 lines 412-415 This paragraph may need references. However it is hardly understandable, please clarify.

Pg 14 lines 417-418 The assumption that RoM does not change in 2 days is not correct. This is a major flaw.

Conclusions

Pg 15 lines 431-432 Authors should explain which kind of validity has been proved.

Finally we should mention a great missing point, responsiveness. With data collected it could have been possible to calculate at least de minimal detectable change. Authors should explain why they did not do that.

Reviewer 3 Report

There have been only 15 people, no need for ICC

The results should be presented with more descriptive variables as ICC is the proper Reliability coefficient for more testers

The conclusion is not supported by the results: a measurement with partially only ICC 0.61 is of no use

The study is not relevant as the gold standard must be used for comparisson Cronbachs Alpha instead of ICC must be used, more testers and more patients should be included for reliability studies

I recommend using this as a pilot study for DTA sample size calculation and publish the results along with final study

Round 2

Reviewer 1 Report

 Accept in present form

Minor editing of English language required

Author Response

Thank you for your valuable feedback on our research publication. We appreciate your assessment and we have taken necessary steps to address the minor editing required for the English language.

To ensure the linguistic accuracy of our paper, we engaged a native English speaker with expertise in our research field to carefully review and edit the manuscript. This linguistic expert made necessary corrections to improve the clarity and coherence of the content.

Furthermore, we utilized advanced language processing tools such as Grammarly, which efficiently identified and rectified any grammatical errors, spelling mistakes, and punctuation issues.

As a result of these efforts, we believe the manuscript is well-crafted and effectively communicates our research findings. The improved language quality contributes to the overall understanding of the study and enhances the reading experience for the audience.

Once again, we express our gratitude for your constructive feedback, which has significantly contributed to the overall improvement of our publication. We hope that the revised version meets the high standards of language proficiency expected for scholarly work.

Reviewer 2 Report

I really appreciate the changes made. However the issue of responsiveness (that is change detection potential) should be adressed before acceptance. I insist that the authors should have the data to at least calculate minimal detectable change. 

Author Response

Thank you for your valuable feedback on our manuscript. We genuinely appreciate your effort in reviewing our work. We have carefully considered your comments and have made the necessary changes accordingly.

Regarding the issue of responsiveness and change detection potential, we completely agree with your suggestion. To address this concern, we conducted a power analysis before conducting the study. We utilized the results published by Rigoni et al. as a basis for our calculations. The assumptions we used for the power analysis were as follows: Effect size = 5 degrees, standard deviation = 5 degrees, alpha = 0.05, and a power of 0.90. Based on these parameters, the minimum required sample size was determined to be n = 26.

Moreover, in response to your recommendation, we performed the calculation for the Minimal Detectable Change (MDC).

Once again, we sincerely appreciate your insightful comments and are grateful for the opportunity to improve the quality of our research.

Reviewer 3 Report

nothing to add.

Author Response

Thank you for taking the time to review our research publication. We appreciate your assessment and are delighted to receive your positive feedback.

Your acknowledgment of the thoroughness and completeness of our manuscript is truly encouraging. We are pleased to know that the content presented meets the required standards and effectively conveys the research findings.